

# DNA barcoding of Austrian snow scorpionflies (Mecoptera, Boreidae) reveals potential cryptic diversity in *Boreus westwoodi*

Lukas Zangl[1,2], Elisabeth Glatzhofer[1], Raphael Schmid[1], Susanne Randolf[3] and Stephan Koblmüller[1]

[1] Institute of Biology, University of Graz, Graz, Styria, Austria
[2] Universalmuseum Joanneum, Studienzentrum Naturkunde, Graz, Styria, Austria
[3] Natural History Museum Vienna, Vienna, Vienna, Austria

## ABSTRACT

**Background**. Snow scorpionflies (genus *Boreus*) belong to a family of Mecoptera, Boreidae, that has been vastly neglected by entomological researchers due to their shift in seasonality to the winter months. Their activity during this time is regarded as a strategy for predator avoidance and regular sightings on snow fields suggest that this also facilitates dispersal. However, many aspects about snow scorpionflies, especially systematics, taxonomy, distribution of species, phylogenetics and phylogeography have remained fairly unexplored until today. In this study, we fill some of these gaps by generating a reference DNA barcode database for Austrian snow scorpionflies in the frame of the Austrian Barcode of Life initiative and by characterising morphological diversity in the study region.

**Methods**. Initial species assignment of all 67 specimens was based on male morphological characters previously reported to differ between *Boreus* species and, for females, the shape of the ovipositor. DNA barcoding of the mitochondrial cytochrome c oxidase subunit 1 (COI) gene was carried out for all 67 samples and served as a basis for BIN assignment, genetic distance calculations, as well as alternative species delimitation analyses (ABGD, GMYC, bGMYC, bPTP) and a statistical parsimony network to infer phylogenetic relationships among individual samples/sampling sites.

**Results**. Morphological investigations suggested the presence of both *Boreus hyemalis* and *Boreus westwoodi* in Austria. DNA barcoding also separated the two species, but resulted in several divergent clades, the paraphyly of *B. westwoodi* in Austria, and high levels of phylogeographic structure on a small geographic scale. Even though the different molecular species delimitation methods disagreed on the exact number of species, they unequivocally suggested the presence of more than the traditionally recognized two *Boreus* species in Austria, thus indicating potential cryptic species within the genus *Boreus* in general and especially in *B. westwoodi*.

Corresponding author
Lukas Zangl, lukas.zangl@uni-graz.at

## INTRODUCTION

The holometabolous insect genus *Boreus* (Mecoptera: Boreidae), commonly known as snow scorpionflies or winter scorpionflies (*Ibrahimi et al., 2018*), has a Holarctic distribution and is most famous for its cold tolerance and activity on snow (*Hågvar & Ostbye, 2011*). The imagines occur from about October to March/April (*Finch, 1997*; *Hågvar, 2001*), a temporal niche, which, among other hypotheses, has been attributed to predator avoidance and improved dispersal (*Hågvar, 2010*). Snow scorpionflies predominantly feed on mosses, but are also known to process decaying insects (*Finch, 1997*). Despite a basic understanding of their general biology (*Finch, 1997*; *Hågvar, 2001*; *Hågvar, 2010*), gaps in the knowledge concerning their distribution and species richness are yet to be overcome (e. g., *Willmann, 1978*; *Hågvar & Ostbye, 2011*; *Ibrahimi et al., 2018*). However, most of the existing literature unanimously reports *Boreus hyemalis* (Linnaeus, 1767) and *Boreus westwoodi* Hagen, 1866 from Southwest Europe to Northeast Scandinavia, and consequently also from Austria (*Willmann, 1978*; *Devetak, 1988*; *Finch, 1997*; *Raemakers & Kleukers, 1999*; *Kreithner, 2001*; *Hågvar & Ostbye, 2011*; *Tillier, Callot & Ragué, 2011*; *Ibrahimi et al., 2018*). Field studies suggested similar ecological preferences for these two species (*Hågvar, 2010* and references therein) and therefore some authors have also regarded them as only one species (e.g., *Saure, 2003*). Other species like *Boreus lokayi* Klapálek 1901 (Romania, Slovakia), *Boreus aktijari* Pliginskij, 1914 (Crimea) or *Boreus kratochvili* (*Mayer, 1938*) (Czech Republic) are only scarcely mentioned in the literature (*Penny, 1977*; *Willmann, 1978*; *Kreithner, 2001*; *Ibrahimi et al., 2018*) and the latter one is even regarded a synonym of *B. hyemalis* (*Kreithner, 2001*). *Boreus gigas* (*Brauer, 1876*) is another ambiguous taxon, which is currently also considered a synonym of *B. hyemalis* and even lacks a formal species description at all (*Willmann, 1978*). In the past, descriptions of *Boreus* species were based exclusively on morphological characters (*Brauer, 1876*; *Mayer, 1938*; *Blades, 2002*). Morphological similarity, plasticity and overlapping ranges, though, have issued continuous discussions about their validity (*Willmann, 1978* and references therein; *Kreithner, 2001*) and consequently the distribution of distinct species across Europe in general (*Willmann, 1978*; *Finch, 1997*; *Kreithner, 2001*), but also for Austria in particular (*Gepp, 1982*; *Kreithner, 2001*; *Gruppe & Aistleitner, 2011*). A detailed morphological study compared material from the Alps (Austria, Switzerland, Slovenia, Italy and France) with specimens from Croatia and Sweden and provided a set of morphological characters for species discrimination, spanning some of the intraspecific and geographic morphological variation (*Kreithner, 2001*). However, no relevant genetic information of European *Boreus* species has been available so far.

Since DNA barcoding was introduced as a method for biological species discrimination (*Hebert, Ratnasingham & De Waard, 2003*), several studies have shown that its delimiting powers also apply to various insect groups (e.g., *Raupach et al., 2016*; *Huemer et al., 2019*; *Zangl et al., 2019*; *Galimberti et al., 2020*). However, DNA barcoding also has well known limitations with respect to recently diverged species, large population sizes retaining divergent haplotypes and hybridization/introgression (e.g., *Van Velzen et al., 2012*; *Ermakov et al., 2015*; *Cong et al., 2017*; *Zangl et al., 2020*; *Paill et al., 2021*), and species

delimitation therefore benefits from additional sources of data (e.g., *Trewick, 2008*; *Liu et al., 2017*). Conducted in the framework of the Austrian Barcode of Life initiative (ABOL, http://www.abol.ac.at; *Haring, Sattmann & Szucsich, 2015*), the present study aims at (i) contributing DNA barcodes of Austrian *Boreus* species to the Barcode of Life database (BOLD; http://www.boldsystems.org; *Ratnasingham & Hebert, 2007*), (ii) investigating their genetic diversity, (iii) validating the two proposed Central European species with genetic data and (iv) testing whether genetic results mirror the morphological variability displayed by both *B. hyemalis* and *B. westwoodi*.

## MATERIALS & METHODS

All specimens investigated in the present study were collected in concordance with state conservation laws and under following permits granted by the Amt der Steiermärkischen Landesregierung, Abt. 13 Umwelt und Raumordnung and the Amt der Kärtner Landesregierung, Abt. 8 Umwelt, Energie und Naturschutz, respectively: ABT13-53S-7/1996-156, ABT13-53W-50/2018-2, 08-NATP-845/1-2019(007/2019), N-2018-326688/8-Pin). From 2017 to 2020, 67 individuals from 18 Central and Eastern Austrian localities were caught by hand and stored in 2 ml Eppendorf tubes in pure Ethanol at $-20$ °C (information on species determinations, collection and storage is available on BOLD (dx.doi.org/10.5883/DS-BOREUS), Table S1). Morphological species discrimination followed *Penny (1977)* and *Kreithner (2001)*. Primarily, the shape of tergal apophyses (TA), gonostyles (GS), epiandrum (EA) and hypandrum (HA) of males was used to assign specimens to species, as these have been identified as the most reliable discriminating characters in previous studies (*Kreithner, 2001* and references therein). Due to the unexpected genetic diversity recovered by the DNA barcoding (see below), further morphological investigation included the properties of the caput, the number of antennal segments and the number of bristles on the front wing. These characters, however, have been regarded as questionable or even unsuited for species discrimination in previous studies (*Kreithner, 2001* and references therein) but we wanted to check if they show any correspondence to the genetic results. For females, *Kreithner (2001)* suggested the shape of the ovipositor and especially of the gonocoxosternites, which we also used as the primary distinctive character. A Keyence digital microscope was used to assess TA, EA and HA in males and to capture the general appearance of all specimens.

For DNA analyses, total genomic DNA was extracted from three legs using the NucleoSpin Tissue XS Micro kit (Macherey-Nagel) following the manufacturer's instructions. PCR amplification, purification and chain termination sequencing using the primer set C_LepFolF and C_LepFolR (*Hernández-Triana et al., 2014*) followed *Koblmüller et al. (2011)* and *Duftner, Koblmüller & Sturmbauer (2005)*. Sequences were visualized on a 3500xl capillary sequencer (ABI) and aligned by Muscle in MEGA 6.06 (*Tamura et al., 2013*). Clustering analysis based on a Neighbor-Joining (NJ) tree was performed using the "Taxon ID Tree" tool implemented on BOLD (http://www.boldsystems.org) based on a muscle alignment and employing the pairwise deletion option. Genetic distances within and between main lineages/species were calculated using the "Barcode Gap

Analysis" tool, also provided on BOLD. Furthermore, we estimated divergence times by translating COI distances under the assumption of a general arthropod divergence rate of 1.0-2.5% per MY (e.g., *Brower, 1994*; *Quek et al., 2004*; *Papadopoulou, Anastasiou & Vogler, 2010*; *Pons et al., 2010*). Sequences of *Boreus borealis* (KU874461.1, KU874462 (*Sikes et al., 2017*)), a North American representative of the genus, were downloaded from GenBank and used as outgroup. For molecular species delimitation, BIN assignment on BOLD (*Ratnasingham & Hebert, 2013*), the Automatic Barcode Gap Discovery (ABGD) (*Puillandre et al., 2011*), the Bayesian Poisson Tree Processes (bPTP) model (*Zhang et al., 2013*), the Generalized Mixed Yule Coalescent (GMYC) (*Zhang et al., 2013*), and the Bayesian GMYC (bGMYC) (*Reid & Carstens, 2012*) were used. ABGD was performed via the web version (https://bioinfo.mnhn.fr/abi/public/abgd/abgdweb.html) using default settings and each of the three distance models Kimura (K80) TS/TV, Jukes-Cantor (JC69) and Simple Distance (results are reported for the Kimura (K80) TS/TV model here as they did not vary between the different models). As input tree for the bPTP analysis, a Maximum Likelihood (ML) tree was inferred via the web-version of PhyML 3.0 (http://www.atgc-montpellier.fr/phyml/; *Guindon et al., 2010*), employing the HKY model (selected by SMS in PhyML based on the Bayesian Information Criterion (BIC); *Lefort, Longueville & Fascuel, 2017*), no preset starting tree and 1000 bootstrap pseudo-replicates to assess nodal support. bPTP was run on the web server (https://species.h-its.org/ptp/) using the default settings (100,000 MCMC generations, thinning = 100, burn-in value = 0.1, seed = 123). For the GMYC analysis, an ultrametric tree was inferred in BEAST v.2.6.3 (*Bouckaert et al., 2019*). The MCMC chain was run for 10 million generations (sampling frequency = 5,000) employing the HKY model, a strict molecular clock and a birth-death tree prior. ESS values (all > 200) were checked with Tracer v1.7 (*Rambaut et al., 2018*). TreeAnnotator v2.6.3 (part of the BEAST2 package) was used to infer a maximum clade credibility tree from the set of posterior trees. GMYC was run on the web server (https://species.h-its.org/gmyc/) with the single threshold option. The bGMYC analysis was conducted on 501 posterior trees from the BEAST analysis and run (MCMC = 50,000; burnin = 40,000; thinning = 100) in R v3.6.0 (*R Core Team, 2013*) using the package bGMYC v.1.0.2 (*Reid & Carstens, 2012*). We used a rather conservative posterior probability threshold (posterior probability: $0.5 < P < 0.9$) to identify putative species, compared to higher thresholds that might overestimate the species' number (*Kornilios et al., 2020*).

Furthermore, we calculated the number of haplotypes (h), the haplotype diversity (Hd) and the nucleotide diversity ($\Pi$) for the whole dataset using DnaSP v6 (*Rozas et al., 2017*). Finally, a statistical maximum parsimony network was inferred using TCS (*Clement et al., 2002*) with default settings as implemented in PopART v.1.7 (*Leigh & Bryant, 2015*) to visualize phylogeographic relationships. The input file was created using a custom-made python script (available on https://github.com/maxwagn/popart_popprep).

## RESULTS

Morphological determination resulted in one *B. hyemalis* and 28 *B. westwoodi* males and one *B. hyemalis* and 37 *B. westwoodi* females (Figs. 1, 2, Table 1).

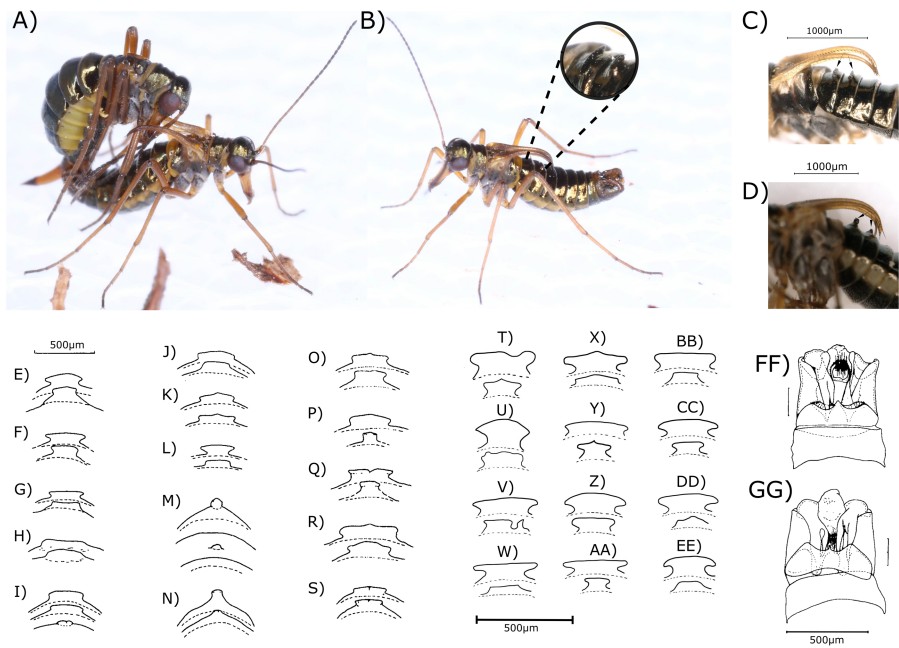

**Figure 1** **Images and drawings of boreids and their morphological characters.** Images of a copulation (A) and a single male (B) *Boreus westwoodi*, as well as digital microscopy images of the anvil-shaped tergal apophyses (TA) of *B. westwoodi* (C) and pointed TA of *B. hyemalis* (D) from Austria (indicated by black arrows). (E–EE) Drawings of types of TA of *B. westwoodi* (forms E–L), *B. hyemalis* (forms M–N) as well as forms of uncertain taxonomic status (O–S) from across Europe (edited from *Kreithner (2001)*) and Austrian *B. westwoodi* (T–EE). Drawings of the main shapes of the genital segments (GS) with the epiandrum (EA) (FF–GG, see Table 1) retrieved and edited from *Kreithner (2001)*. ©Photos by Elisabeth Glatzhofer. ©Nikolaus Romani.

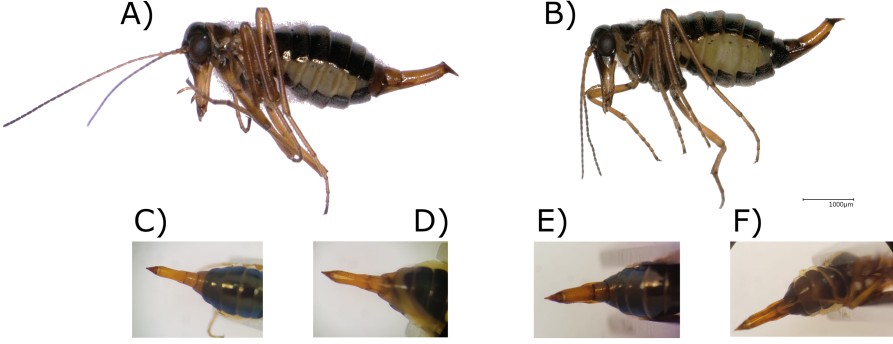

**Figure 2** **Images of *Boreus* females and their ovipositors.** Digital microscopy images of (A) *Boreus westwoodi* and (B) *B. hyemalis* females as well as the dorsal (C and E) and ventral (D and F) view of their ovipositor, respectively.

The investigated morphological characters in males showed different degrees of variation between the specimens. TAs, which have been regarded as reliable for species discrimination, were anvil-shaped in all except one (294) *Boreus* males, resembling one

**Table 1  Morphological characterization of Austrian male *Boreus* spp. according to *Kreithner (2001)*.** Numbers in Field ID correspond with Table S1. Tergal apophyses (TA), epiandrum (EA), gonostylus (GS). Numbers in GS correspond to figures in *Kreithner (2001)*. Forms of TA correspond with Figs. 1T–1EE.

| Field ID | Caput | No. of antennal segments | TA | No. of bristles on outer side of front wing (external/internal) | EA | GS |
|---|---|---|---|---|---|---|
| 200 | Corrugated | 24 | Form Z | 8/36 | Lateral lobes shorter than septum, septum broad triangular | 41 |
| 208 | Corrugated | 25 | Form T | 2/33 | Lateral lobes shorter than septum, septum broad triangular | 41 |
| 209 | Corrugated | 24 | Form U | 9/31 | Lateral lobes same length as septum, septum broad triangular | 41 |
| 210 | Corrugated | 25 | Form W | 9/10 | Lateral lobes shorter than septum, septum broad triangular | 41 |
| 211 | Corrugated | 23 | Form Z | 9/29 | Lateral lobes shorter than septum, septum broad triangular | 41 |
| 212 | Corrugated | 24 | Form Z | 10/30 | Lateral lobes shorter than septum | 41 |
| 214 | Corrugated | 23 | Form V | 10/33 | Lateral lobes shorter than septum, septum broad triangular | 41 |
| 215 | Corrugated | 23 | Form Z | 6/23 | Lateral lobes shorter than septum, septum broad triangular | 41 |
| 216 | Corrugated | 16 | Form X | 2/33 | Lateral lobes shorter than septum, septum broad triangular | 41 |
| 265 | Corrugated | 25 | Form AA | 11/33 | Lateral lobes shorter than septum, septum broad triangular | 41 |
| 266 | Smooth, pilose | n.a. | Form AA | 9/28 | Lateral lobes shorter than septum, flat, septum broad triangular | 41 |
| 267 | Corrugated | 23 | Form AA | 7/29 | Lateral lobes shorter than septum, flat, septum broad triangular | 41 |
| 268 | Corrugated | 25 | Form AA | 13/36 | Lateral lobes shorter than septum, flat, septum broad triangular | 41 |
| 269 | Corrugated | 23 | Form AA | 8/30 | Lateral lobes shorter than septum, flat, septum broad triangular | 41 |
| 274 | Corrugated | 25 | Form CC | 12/32 | Lateral lobes shorter than septum, septum broad triangular | 41 |
| 275 | Corrugated | 24 | Form AA | 9/32 | Lateral lobes longer than septum, septum broad triangular | 41 |
| 276 | Corrugated | 23 | Form AA | 13/31 | Lateral lobes shorter than septum, septum broad triangular | 41 |
| 277 | Corrugated | 24 | Form X | 11/26 | Lateral lobes longer than septum, septum broad triangular | 41 |
| 278 | Corrugated | 23 | Form BB | 8/20 | Lateral lobes shorter than septum, septum broad triangular | 41 |
| 280 | Corrugated | 24 | Form CC | 8/34 | Lateral lobes same length as septum, septum broad triangular | 41 |
| 282 | Corrugated | 24 | Form X | 10/32 | Lateral lobes same length as septum, septum broad triangular | 41 |

| Field ID | Caput | No. of antennal segments | TA | No. of bristles on outer side of front wing (external/internal) | EA | GS |
|---|---|---|---|---|---|---|
| 284 | Corrugated | 24 | Form EE | 10/34 | Lateral lobes same length as septum, septum broad triangular | 41 |
| 285 | Corrugated | n.a. | Form EE | 8/25 | Lateral lobes same length as septum, septum broad triangular | 41 |
| 287 | Corrugated | n.a. | Form EE | 8/32 | Lateral lobes shorter than septum, flat, septum broad triangular | 41 |
| 290 | Corrugated | n.a. | Form DD | 8/24 | Lateral lobes shorter than septum, septum broad triangular | 41 |
| 291 | Corrugated | 24 | Form DD | 8/27 | Lateral lobes same length as septum, septum broad triangular | 41 |
| 292 | Corrugated | n.a. | Form W | 8/27 | Lateral lobes shorter than septum, septum broad triangular | 41 |
| 294 | Corrugated | 23 | Form U | 2/33 Hind wings with 2 diffuse rows of 23 bristles | Lateral lobes longer than septum, septum pointed | 45 |
| 296 | Corrugated | 23 | Form AA | 11/26 | Lateral lobes shorter than septum, septum broad triangular | 41 |
| 297 | Corrugated | 23 | Form AA | 14/32 | Lateral lobes shorter than septum, septum broad triangular | 41 |
| 299 | Corrugated | 24 | Form Z | 8/33 | Lateral lobes same length as septum, flat, septum broad triangular | 41 |
| 301 | Corrugated | 25 | Form X | 10/39 | Lateral lobes shorter than septum, septum broad triangular | 41 |
| 303 | Corrugated | 24 | Form CC | 12/39 | Lateral lobes shorter than septum, septum broad triangular | 41 |
| 307 | Corrugated | 24 | Form CC | 10/34 | Lateral lobes shorter than septum, septum broad triangular | 41 |
| 308 | Corrugated | 24 | Form BB | 12/23 | Lateral lobes shorter than septum, septum broad triangular | 41 |
| 312 | Corrugated | 25 | Form Y | 10/34 | Lateral lobes shorter than septum, septum broad triangular | 41 |
| 314 | Corrugated | 25 | Form BB | 11/34 | Lateral lobes shorter than septum, septum broad triangular | 41 |
| 318 | Corrugated | 24 | Form CC | 13/39 | Lateral lobes shorter than septum, septum broad triangular | 41 |

of the 12 forms presented in Figs. 1T–1EE. Among the different anvil shapes, little to no geographical pattern became obvious as for example all males from Hochwechsel share the same TA shape, but this particular shape was also recovered from males from Weinebene and Gösting. Some of the shapes recovered from Austrian samples also resembled the forms 35–39 (Figs. 1O–1S), which *Kreithner (2001)* referred to as taxonomically uncertain. Similar results were also obtained for the shape of the GS, which generally appeared much more slender in *B. westwoodi*. Only the *B. hyemalis* male (294) showed a distinctly different form of the GS with a bulkier appearance and a longer medituberculus (Table 1), all *B. westwoodi* males regardless of their sampling location shared the same GS shape. The number of antennal segments, the number of bristles on the outer side of the front

wing and the shape of the EA, all considered as unreliable characters (*Kreithner, 2001*), did neither correlate with the two species nor with geographical origins. Characters of the ovipositor like the ventral membranous part or the lateral tapering were remarkably homogenous among all *B. westwoodi* females and similar to *B. hyemalis*. The only difference between *B. westwoodi* and *B. hyemalis* females was the breadth of the proximal part of the gonocoxosternites, which was broader in *B. hyemalis*. However, since we only have one nominal *B. hyemalis* female included in our study, we are cautious to rely on this character as we cannot estimate the extent of plasticity.

DNA barcodes of the partial COI gene ranging from 649 to 657 bp in length were generated for 67 specimens (sequences are available on BOLD (dx.doi.org/10.5883/DS-BOREUS) and GenBank (MW627590–MW627656). These sequences were grouped into seven BINs (Figs. 3 and 4), four of which were newly created (BOLD:AEF6177, BOLD:AEF6178, BOLD:AEF6179 and BOLD:AEF8503). The different BINs, however, cannot be distinguished morphologically from each other, except for the BOLD:ACT2769, which comprises *B. hyemalis*. Based on the results of the DNA barcoding, *Boreus westwoodi* turned out to be paraphyletic with respect to *B. hyemalis* (Fig. 3, Fig. S1). As we only had two *B. hyemalis*, the actual extent of intraspecific K2P distance within this species cannot be discussed here. However, since both specimens were from the same sampling site, the observed intraspecific distance was expectedly low. Intraspecific distances within *B. westwoodi* (up to 7.21%) in part considerably exceeded interspecific distances (2.53–5.9%). Species delimitation analysis results differed considerably among the four alternative methods. While ABGD inferred four species in the recursive approach including the outgroup (three species in the initial approach, Fig. S2) and GMYC suggested six species, bPTP estimated 19 to 44 (mean 28) and bGMYC resulted in 10 species, when assuming a threshold $0.5 < P < 0.9$ for conspecificity (Fig. 3). Overall, we found 55 haplotypes (h) across the whole dataset (Hd = 0.99394, Π = 0.03362).

The statistical maximum parsimony network revealed some phylogeographic structure and very little haplotype sharing among and even within sampling sites. Even though geographically close sampling sites often group together in the network, there are some exceptions to this pattern (Fig. 5). Thus, the samples from Hochwechsel resulted in a part of the network (and tree, Fig. 3) that otherwise comprises samples collected west of the Mur River. On the other hand, all samples from Gösting, which is geographically close to Thal (distance < 3 km) and west of the Mur River, grouped with samples north and east of the Mur River, even though they formed a quite distinct cluster there. Interestingly, the *B. westwoodi* lineage most closely related to *B. hyemalis* showed a pattern different from the rest of *B. westwoodi*, i.e., haplotype sharing was found among geographically distant sampling sites and the overall genetic diversity in this clade was low.

## DISCUSSION

In this study, we provide 67 new DNA barcodes representing the first genetic insights into the snow scorpionfly diversity of the genus *Boreus* from Austria and thus also Europe. This apparent lack of genetic information may be attributed to a certain characteristic of the
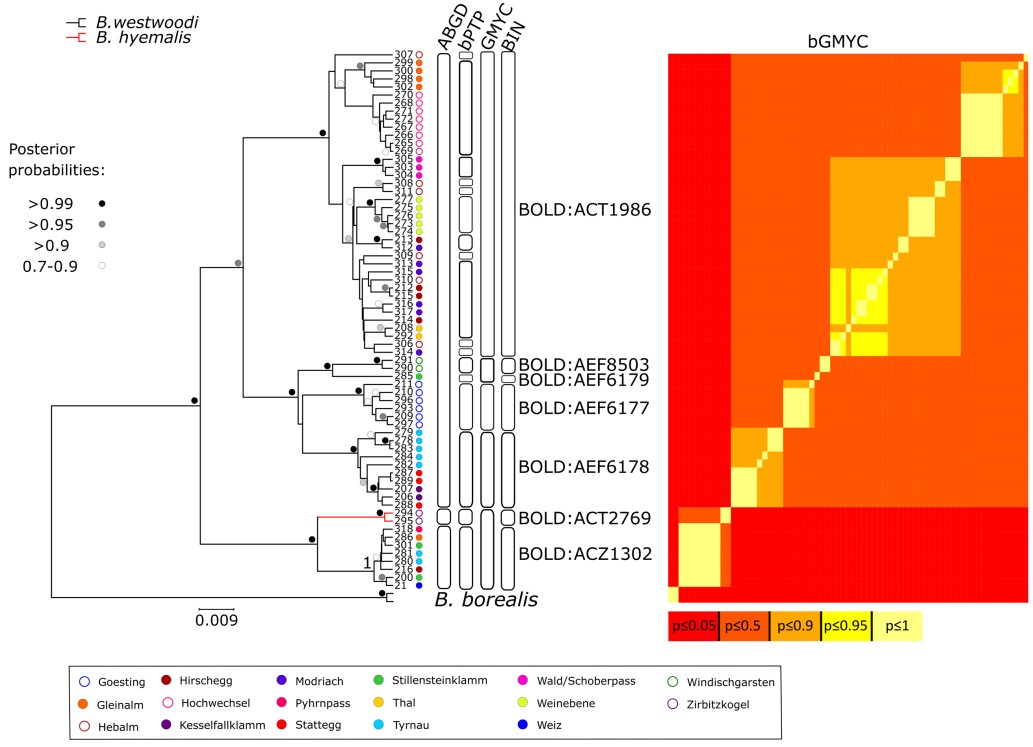

**Figure 3 Bayesian inference phylogeny and species delimiation.** Bayesian Inference (BI) phylogeny based on the DNA barcoding region (part of the mitochondrial COI gene). Tip numbers correspond with Table S1 and represent specimen IDs. Colored dots represent sample origin. Colored branches indicate initial morphological species assignment. Dots near nodes represent posterior probability categories. Boxes and heatmap to the right indicate the number of putative species inferred by different molecular species delimitation methods.

boreids' biology. Due to their shift in seasonality of the imaginal stage to the winter months, few entomologists ever collect them as bycatch from passive stationary traps, let alone actively pursue them (*Hågvar & Ostbye, 2011*). Consequently, contemporary literature about Boreidae almost exclusively only covers new records (*Tillier, Callot & Ragué, 2011*; *Ibrahimi et al., 2018*), re-evaluates national distribution of species (*Devetak, 1988*; *Finch, 1997*; *Raemakers & Kleukers, 1999*; *Tillier, Callot & Ragué, 2011*; *Hågvar & Ostbye, 2011*) and conducts morphological comparison of already available material (*Kreithner, 2001*). However, phenotypic plasticity has been found to be extensive both within species and across larger geographic distances and has fueled debates about the validity and exact distribution of extant species (*Willmann, 1978*; *Kreithner, 2001*). Nonetheless, certain morphological traits have been reported to hold sufficient discriminative power and suggested the presence of *Boreus hyemalis* and *Boreus westwoodi* throughout Central Europe (*Kreithner, 2001*; *Hågvar & Ostbye, 2011*; *Ibrahimi et al., 2018*) and consequently also in Austria. However, examination of these characters on material from Austria also recovered a high degree of morphological variation at least within *B. westwoodi* (Fig. 1, Table 1). Comparison of the Austrian material with morphological characteristics reported

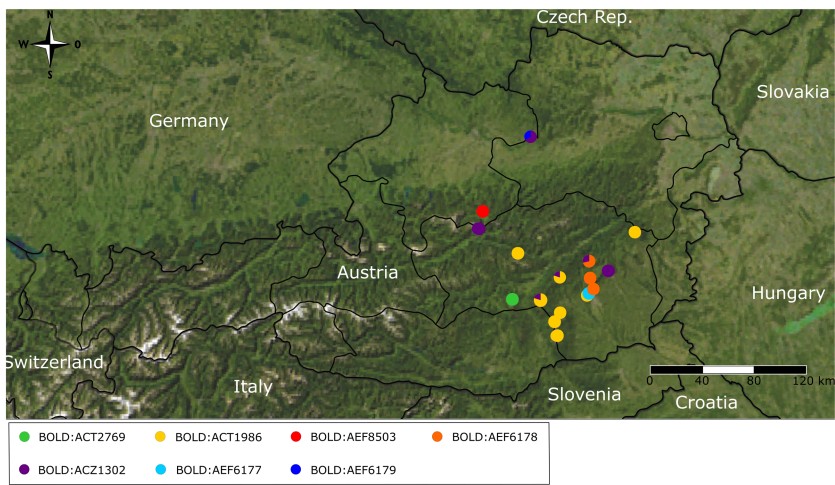

**Figure 4** **Map of geographic BIN distribution.** Distribution map of Barcode Index Number (BIN) composition across Austrian sampling localities.

for European *Boreus* by *Kreithner (2001)* showed that some of the different morphotypes are very similar between the Austrian and the European samples, indicating a large diversity even within Austria. Furthermore, the resemblance of some of the Austrian TA shapes with the forms regarded as taxonomically uncertain by *Kreithner (2001)* potentially hints at the existence of cryptic species. The morphological variation, though, is not perfectly consistent with the distribution of genetic haplotypes as samples from the same location might share similar DNA barcodes but show different morphologies or vice versa (Table 1, Figs. 1 and 5), similar to patterns previously reported for some other arthropods, such as scorpions of the genus *Buthus* in the Atlas Mountains or North African darkling beetles (*Habel et al., 2012*; *Husemann et al., 2012*; *Rangel López et al., 2018*). Since only one single male (sample 294) could be assigned to *B. hyemalis* based on synoptic inspection of all morphological characters, phenotypic plasticity cannot be evaluated here. However, the shape of TA recovered for sample 294 matches form 34 of *Kreithner (2001)* almost perfectly (Fig. 1N) and for the first time links this particular morphotype with a DNA barcode and a particular BIN (BOLD:ACT2769). In females, morphological variability, i.e., ovipositor shape, was virtually non-existent. The sole exception was the female individual collected at Zirbitzkogel, the locality where we also collected a *B. hyemalis* male. Though morphologically similar to other females, this individual had a broader ovipositor than other specimens and, unlike any other specimen, lateral extensions at the proximal part of the gonocoxosternites (compare with *Kreithner, 2001*). These few findings of *B. hyemalis* are also in line with *Kreithner (2001)*, who reported *Boreus* populations across Austria being predominantly *B. westwoodi* with only a few reports of *B. hyemalis* from Eastern Austria. *Kreithner (2001)* also indicated, that reports of sympatric occurrences may reflect cases of misidentification and that additional species could be present, e.g., in alpine regions.

Furthermore, the results of the DNA barcoding and species delimitation analyses suggest that there might be more than two species of *Boreus* present in Austria (Fig. 3). As we had

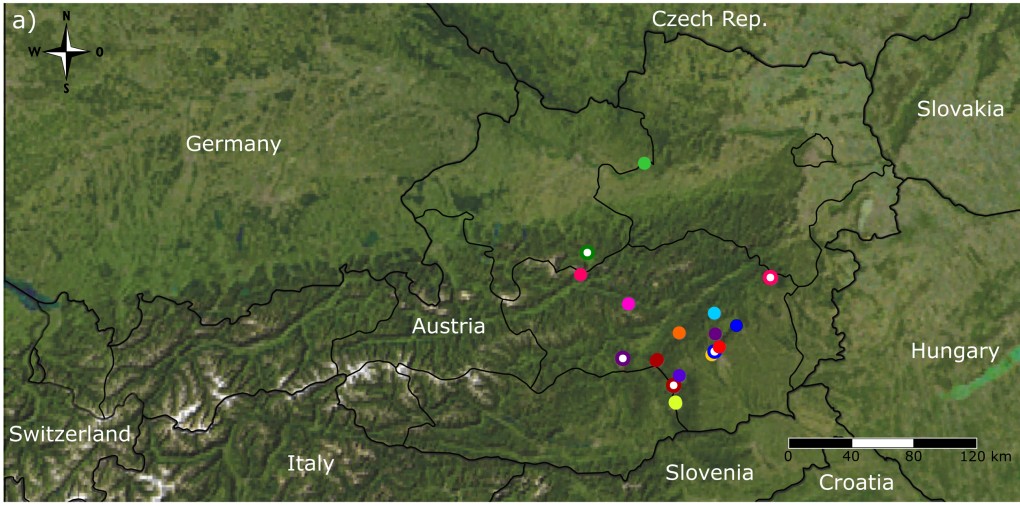

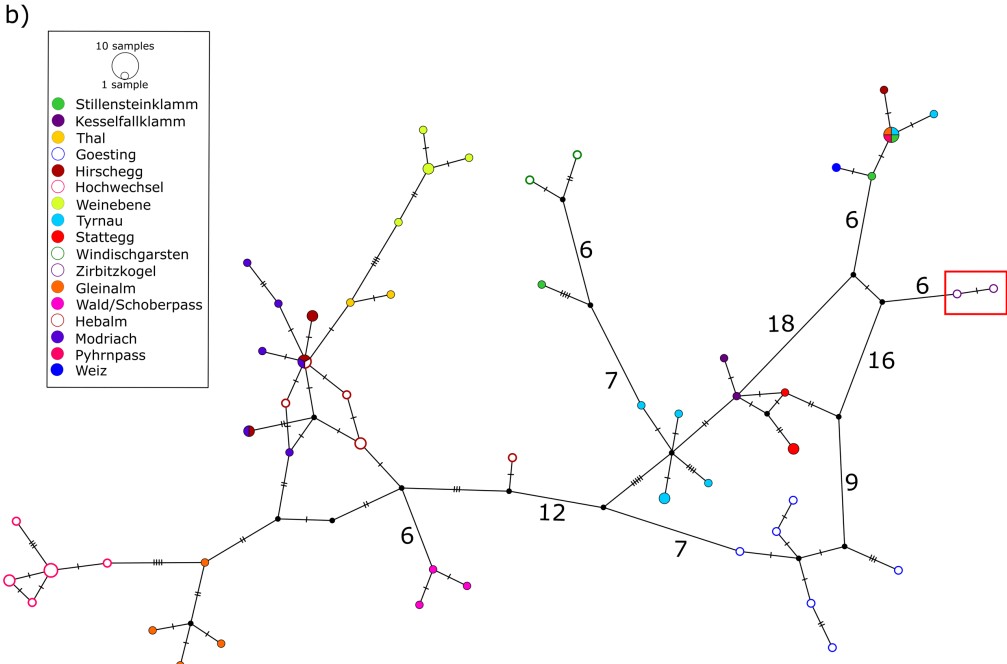

**Figure 5 Sampling map and maximum parsimony network.** Map of Austria and surrounding countries including sampling localities (color coding matches insert in 5B). (B) Statistical parsimony network based on COI sequences. Colors indicate the different sampling localities. Each circle corresponds to one haplotype and its size is proportional to its frequency. Single mutational steps up to five substitutions between haplotypes are indicated by black bars (substitutions > 5 are represented by numbers next to connective lines). Red frame outlines *Boreus hyemalis* specimens.

only two nominal specimens of *B. hyemalis* included in our dataset, no inferences can be drawn about intraspecific genetic diversity or the presence/absence of a barcoding gap. In *B. westwoodi*, maximum intraspecific distances are significantly higher than the distance to their nearest neighbor (Table 2), which has also been reported for e.g., ground beetles,

**Table 2 Genetic distances of *Boreus* spp. based on COI sequences.** Maximum intraspecific K2P distances ($I_{max}$) and distances (DNN distance to nearest neighbor) to nearest neighbor (NN) are listed.

| Species | $I_{max}$ | DNN | NN |
| --- | --- | --- | --- |
| *Boreus hyemalis* | 0.15 | 2.53 | *Boreus westwoodi* |
| *Boreus westwoodi* | 7.21 | 2.53 | *Boreus hyemalis* |

butterflies and aphids (*Raupach et al., 2016*; *Janzen et al., 2017*; *Lee et al., 2017*) and may be an indication for cryptic diversity. Furthermore, interspecific distances of two to three percent separating good species have previously been found in e.g., aphids and mosquitos as well (*Lee et al., 2017*; *Wang et al., 2012*). While distance-based species delimitation methods like ABGD are prone to lump many species together (*da Silva et al., 2018*; *Dellicour & Flot, 2018*; *Galimberti et al., 2020*) and *Dellicour and Flot (2015)* even report ABGD and GMYC as unable to correctly delimit species in scenarios involving only one or two species, ABGD suggested one additional species for Austrian boreids. Tree-based methods on the other hand are known to have a tendency for oversplitting, both in cases with few and many species (*Dellicour & Flot, 2018*), which would explain the large number of species suggested by bGMYC and bPTP in our case. However, despite a general incongruence and a large range in the number of potentially recovered species with the different methods (Fig. 3), they all concur in suggesting that more than the two previously reported species exist in Austria. Limitations for species delimitation inferences based on a single gene are obvious and well discussed in the literature but still can pinpoint ambiguous cases (*Da Silva et al., 2018*; *Galimberti et al., 2020*). However, in the case of Austrian *Boreus*, the patterns obtained from DNA barcoding, species delimitation, statistical maximum parsimony network and morphological analyses do indicate potential cryptic diversity.

Besides potential cryptic diversity, we found a remarkable geographic structure, with distinct haplogroups present in geographically close populations, sometimes only a few kilometers apart. Even though our sample comprises only a few animals per location, the general lack of haplotype sharing among most sampling sites (with a few exceptions) is striking, and the large number of singletons indicates large (effective) population sizes. Assuming a general range of arthropod COI divergence rates of 1.0–2.5% per MY (e.g., *Brower, 1994*; *Quek et al., 2004*; *Papadopoulou, Anastasiou & Vogler, 2010*; *Pons et al., 2010*), the observed pairwise distances among main lineages of ~2.5–7.2% translate into divergence times of ~1–2.5 to 2.9–7.2 MY. As *Boreus* spp. are flightless, some phylogeographic structure was expected, even considering the rather small geographic scope of our study. Yet, the extent of structure is surprising and unexpected, even though in general, flightless and/or less mobile taxa show higher levels of population genetic differentiation than good dispersers (*Papadopoulou et al., 2009*). Previous studies on small flightless arthropods with alleged low dispersal ability found varying patterns, from little phylogeographic structure with haplotype sharing across long distances (e.g., oribatid mites of the genus *Cymbaeremaeus*, *Schäffer, Kerschbaumer & Koblmüller, 2019*) to deeply divergent genetic lineages without gene flow, potentially representing cryptic species, across distances of only tens of kilometers (e.g., springtails of the genus *Lepidocyrtus*, *Ciccionardi*

*et al., 2010*). The patterns observed in Austrian *Boreus* fit this latter extreme (disregarding the *B. westwoodi* haplogroup most closely related to *B. hyemalis* that was shared among some samples from distant localities). In addition, the factors and processes underlying the peculiar phylogeographic pattern observed in Austrian *Boreus* remain unclear, but it appears that *Boreus* are indeed rather stationary and do not generally disperse over larger distances.

## CONCLUSIONS

In conclusion, this study presents the first genetic information on the genus *Boreus* in Austria, and consequently also Europe. Furthermore, it provides several new localities from which boreids have not been reported within Austria so far and thus augments their known distribution range in Austria. DNA barcodes linked to different morphotypes prove the presence of *Boreus westwoodi* and *Boreus hyemalis* in Austria, high levels of phylogeographic structure on small geographic scales, and indicate the potential presence of further cryptic species. The phenotypic plasticity previously reported for these two species is confirmed in the present study as several morphological characters show a large variation that does not correlate with genetic variation. Seven distinct BINs were recovered by BOLD and several, albeit inconsistent, potential species were suggested across the alternative species delimitation analyses. This potential cryptic diversity probably also extends to other European populations of *Boreus* but disentangling the exact number of species, possible (ancient) hybridization/introgression, (lack of) gene flow among localities and the precise distribution of these species will require further multilocus or genomic as well as morphological investigations and a pan-European sampling of boreids.

## ACKNOWLEDGEMENTS

We are grateful to Lukas Strohmaier, Barbara Bernhart, Thomas Bernhart and Christian Komposch for their help in sample collection. Furthermore, we would like to thank Martin Husemann and two other anonymous reviewers for their constructive comments which helped to improve this manuscript.

### Funding

This research was funded by the Austrian Federal Ministry of Science, Research and Economy in the frame of an ABOL associated project within the framework of the "Hochschulraum-Strukturmittel" Funds. Financial support was also provided by the University of Graz for covering the costs for open access publication. The funders had no role in study design, data collection and analysis, decision to publish, or preparation of the manuscript.

### Grant Disclosures

The following grant information was disclosed by the authors:

Austrian Federal Ministry of Science, Research and Economy in the frame of an ABOL associated project within the framework of the "Hochschulraum-Strukturmittel" Funds. "Hochschulraum-Strukturmittel" Funds.

## Competing Interests

The authors declare there are no competing interests.

## Author Contributions

- Lukas Zangl conceived and designed the experiments, performed the experiments, analyzed the data, prepared figures and/or tables, authored or reviewed drafts of the paper, and approved the final draft.
- Elisabeth Glatzhofer performed the experiments, analyzed the data, prepared figures and/or tables, authored or reviewed drafts of the paper, and approved the final draft.
- Raphael Schmid performed the experiments, analyzed the data, authored or reviewed drafts of the paper, and approved the final draft.
- Susanne Randolf conceived and designed the experiments, authored or reviewed drafts of the paper, and approved the final draft.
- Stephan Koblmüller conceived and designed the experiments, prepared figures and/or tables, authored or reviewed drafts of the paper, and approved the final draft.

## Field Study Permissions

The following information was supplied relating to field study approvals (i.e., approving body and any reference numbers):

All specimens investigated in the present study were collected in concordance with state conservation laws and under following permits granted by Amt der Steiermärkischen Landesregierung, Abt. 13 Umwelt und Raumordnung and Amt der Kärtner Landesregierung, Abt. 8 Umwelt, Energie und Naturschutz [ABT13-53S-7/1996-156, ABT13-53W-50/2018-2, 08-NATP-845/1-2019(007/2019), N-2018-326688/8-Pin)].

## DNA Deposition

The following information was supplied regarding the deposition of DNA sequences:

All sequences (AMEC021-20 to AMEC318-20) are publicly available at BOLD: dx.doi.org/10.5883/DS-BOREUS.

They are also available at GenBank: MW627590–MW627656.

## Data Availability

The COI sequences are available in the Supplemental File.

The input file was created using a custom-made Python script available on GitHub at https://github.com/maxwagn/popart_popprep.

## Supplemental Information

Supplemental information for this article can be found online at http://dx.doi.org/10.7717/peerj.11424#supplemental-information.

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
