# Peer review of "DNA barcoding of Austrian snow scorpionflies (Mecoptera, Boreidae) reveals potential cryptic diversity in Boreus westwoodi"

_PeerJ, doi:10.7717/peerj.11424_

## Round 0.1 · original submission · Major Revisions

Dear Dr. Zangl and colleagues:

Thanks for submitting your manuscript to PeerJ. I have now received three independent reviews of your work, and as you will see, the reviewers raised some concerns about the research. Despite this, these reviewers are optimistic about your work and the potential impact it will have on research studying snow scorpionfly phylogenetics and evolution. Thus, I encourage you to revise your manuscript, accordingly, taking into account all of the concerns raised by all three reviewers.

Please provide a map showing the geographic distribution of your variants.

Please ensure that your figures and tables contain all of the information that is necessary to support your findings and observations. Revise incorrect information. The Materials and Methods appear to be missing important information, please try to be more descriptive. All statistical methods should be adequately described such that they are repeatable.

Support for your phylogeny estimations should be provided, as well as branch lengths if drawn as phylograms.

I agree with the concerns of the reviewers, and thus feel that their suggestions should be adequately addressed before moving forward.

I look forward to seeing your revision, and thanks again for submitting your work to PeerJ.

Good luck with your revision,

-joe

·

Basic reporting

The manuscript is clearly written, the used of language is good and easy to understand. The background is well described and the current knowledge is well summarized. The gap in knowledge and the aim of the study is well identified. The figures are of high quality. I would just like to see an additional figure, a map displaying the distribution of genetic variants. The data is all publicly available (after acceptance).

Experimental design

The data presented is new. The data is suitable to answer the posed questions. The aim of the study is meaningful, as the authors study a widely unexplored taxon with a unique and interesting ecology.
The analyses are performed well and represent the state-of-the art. Some of the methods could be described in a bit more detail. Similarly, the results would benefit from some additional details, such as general parameters of genetic diversity in the dataset.

Validity of the findings

The findings are solid and represent important new data for an unexplored taxon. All interpretation is sound and includes not too much speculation. Potential weaknesses are well identified and addressed.

Additional comments

The manuscript is a nice descriptive study that provides important data for an unexplored taxon.

I have just few additional minor suggestions (besides the additional figure above).

Line 35: I would avoid the verb proved, but would rather say suggested.
Line 104: Name of the used kit spelled wrong.
Line 214: Maybe provide the ABGD graph to display the barcode gap.

Discussion
The pattern observed is interesting and very similar to what is seen in sedentary scorpions and darkling beetles in the Atlas, where it is aside from philopatry attributed to the mountainous terrain. May this be an explanation here as well?

Reviewer 2 ·

Basic reporting

Boreidae is an interesting group that the adults normally reach sexual maturity in the Winter, and copulate on the snow, so that the studies on this group are quite limited. This manuscript provides a species delimitation analysis in Boreus westwoodi using DNA barcoding techniques. In general, the paper is well written and can be published in PeeJ.

Experimental design

The authors used the popular DNA Barcoding to resolve the species problem of Austria Boreus. The experimental desighn is normally right.

Validity of the findings

The findings are interesting, and the conclusions are robust.

Additional comments

Boreidae is an interesting group that the adults normally reach sexual maturity in the Winter, and copulate on the snow, so that the studies on this group are quite limited. This manuscript provides a species delimitation analysis in Boreus westwoodi using DNA barcoding techniques. In general, the paper is well written and can be published in PeeJ. However, there are some major concerns that have to be addressed before publication:
Introduction:
Line 44: change "(Insecta: Boreidae)" to "(Mecoptera: Boreidae)"
Line 83: please insert a comma "," before "the present study"
Line 87: change “iv) to test" to "iv) testing...".
Materials and Methods:
Line 103: "For DNA analyses...." should be in a separate paragraph.
Line 106: "the primer combinations" to " the primer set"
Line 108: What algorithm did the authors use to align sequences in MEGA? Clustalx or Muscle? The authors should give more details.
Line 120: Does "10 Mio" mean "10 million generation"? If so, please correct it.
Line 119-122: Why did the authors use the HKY model and a strict molecular clock (clock rate=1.0)?
Line 136-137: change "as implemented in PopART v.1.7...." to "Phylogeographic relationships were visulized using PopART"
Results:
Line 159: "six distinct clusters and one singleton" is not very clear, please mark them in the tree.
Line 163: "As we only had two B. hyemalis,...." , the sample size is too low to make a persuasive conclusion. I suggest to increase the sample size of B. hyemalis from different localities.
Line 165: "Interspecific distances" should be calculated with more species of Boreus added. It seems insufficient to sample only two species, B. westwoodi and B. hyemalis.
Line 167: How did the authors reconstruct the NJ tree? The authors should clarify it in the Materials and Methods. Besides, "NJ tree" and "ML tree" should be in full name because the authors have never mentioned them before.
Line 171: "for conspecifity" or "for conspecificity"?
Line 173-174: change "phylogeographic substructuring" to ""phylogeographic structure", and "sharing" to "shared".
Line 180: "the B. westwoodi clade most closely related to B. hyemalis" is not right. According to the topology of the phylogenetic tree in Fig. 2, B. westoodi is apparently a paraphyletic grade, not a monophyletic clade.
Line 181-182: change "different to" to "different from", and "haplotype sharing" to "the shared haplotype".
Discussion:
Line 237: The authors claimed "we found a remarkable geographic structure", but we can’t see it clearly in the network. If the authors want to clarify geographic structure, please give details about the haplotype diversity (Hd), nuleotide diversity (Nh), genetic differentiation (Fst) using DNAsp, Arlequin, and SAMOVA, etc.
Line 243: How did the authors translate the pairwise distances into divergence times? Did the authors estimate the divergence time based on COI? If so, please show it in the supplementary materials.
L249: change "large distance”to "long distance"

Reviewer 3 ·

Basic reporting

1. The authors incorrectly quoted the Figure 1 (e) from Kreithner (2001). According to the original reference, only 33 and 34 are represented TAs of Boreus hyemalis, and 35-39 were those of taxonomically uncertain forms (35-38: Form I, 39: Form II).

2. The legend of Figure 1 seems to have errors or conflicts. There are two legends for C-D, "schematic drawings and images of TA and GS" and "Digital microscopy images of TA". In addition, labels for each figure are (A), (B) ... in legends but a), b) ... in figures. Arrows shown TA should be added into the microscopy images (C) and (D). Please revise the captions.

3. In Introduction, the authors do not propose some basic information, such as the number of described species in Europe, and a species list known from the Central Europe. In addition, the current situation of elucidation of Austrian Boreus fauna is not fully introduced, so readers cannot clearly understand which record of which species is new information for Austrian fauna. Please consider to add these information.

4. The reference format is inconsistent. For example, full titles and acronyms of journal titles are mixed; only two references have DOI, and one is represented by DOI itself but another is quoted as URL. A thorough review is needed.

5. In Introduction, about the relationship of Boreus hyemalis and B. westwoodi, there is another opinion that two species are conspecific (Sauri, 2003, Entomologische Nachrichten und Berichte 8; I do not check the original resource but some websites such as https://www.insekten-sachsen.de/Pages/TaxonomyBrowser.aspx?id=223691 follow this opinion). I recommend the authors to insert a short comment about this in the historical review.

Experimental design

6. In Table 1, the authors represented the GS shapes by the numbers "41" or "45", which is the figure number in Kreithner (2001), quoting Kreithner's drawing of GS in Figure 1 (G) and (H). But, they do not provide any mention on specific differences in shapes which distinguish from one another; they only wrote as "distinctly different form" (Line 152). The detailed differences should be described so as to replicate the analysis.

7. A threshold value to show the branch support value (>0.95?) and a scale for branch length are not provided in Figure 2. In addition, Is the branch support value not Bayesian posterior probability but maximum likelihood bootstrap for ML tree?

8. I do not understand the reason why the authors chose six characters (TA, EA, HA, caput, antenna and front wing bristles: Line 97-100). Were the former three characters used as important diagnoses in previous taxonomic papers? And why they added latter three characters? Please add some information about the strategy on selection of morphological characters.

9. The authors identified the females based on the geographic location (Line 100-102) but they do not mention the validity. Why the authors did not adopt the ovipositor as a diagnosis of female written in Kreithner (2001)? Why they regard the location as a key for identification? In other words, are the Boreus species show exclusive allopatric distribution pattern or separated emergence season? I recommend the authors to add short comment about this issue.

10. For phylogenetic analysis, they do not provide any information to judge the validity of Boreus borealis as outgroup (Line 111-113). What kind of species is this, and how does this species relate to the target Austrian populations? I recommend the authors to add short comment about this issue.

11. About the records of Boreus hyemalis, one male and one female were recorded in manuscript (Line 142-144), but two males are shown in supplementary table (Panorpa294, 295). Please check which is correct.

Validity of the findings

12. Related to comment 1, in results the authors said "a distinction between B. westwoodi anvil shapes and B. 150 hyemalis anvil shapes like Kreithner (2001) deduced (Fig. 1e, 25-32 for B. westwoodi; 33-39 for 151 B. hyemalis), was not possible" but this statement is due to misinterpreation (33-34: B. hyemalis, 35-38: Form I, 39: Form II). Comparing Figure 1 (E) and 1 (F), TAs of some Austrian individuals seem to be more similar to those of Form I or II than those of B. westwoodi (e.g. f6 and e38). How the authors regard that TAs of all Austrian specimens except B. hyemalis as that of B. westwoodi? Anyway, discussion on the comparison of Kreithner's forms should be added.

13. Related to comment 8, I recommend the authors to clearly state, 1) characters which have been regarded as good species diagnoses in previous papers, 2) characters from which the authors found unrecognized large variation. This helps readers to understand new findings of this research.

14. Please note if any BIN clusters distinguish from remaining ones morphologically.

15. An addition of a map showing geographic distribution of BIN clusters is welcome to understand the complicated situation. Please note if there are any differences of habitat preferences between species and/or BIN clusters.

Additional comments

In summary, the cryptic diversity of this attractive and wingless insect group which the authors reported is an interesting phenomenon, and this is included in the scope of PeerJ journal. On the other hand, there remains issues and errors on the manuscript, and significant revision is required.

---

## Round 0.2 · Minor Revisions

Dear Dr. Zangl and colleagues:

Thanks for revising your manuscript. The reviewers are generally satisfied with your revision (as am I). Great! However, there are a few concerns to address. Please attend to these issues ASAP so we may move towards acceptance of your work.

Best,

-joe

Reviewer 2 ·

Basic reporting

No comments

Experimental design

No comments.

Validity of the findings

No comments.

Additional comments

The revised manuscript has been much improved. However, the reviewer has still the following concerns, and should be happy to recommend it to be published after a minor revision.
1. The insect mutation rate of mtCOI is commonly considered as 2.3% per Ma (i.e., 0.0115 substitutions/site per lineage). We recommend the authors estimate the divergence time based on Brower (1994). Please see the references Brower (1994) "Rapid morphological radiation and convergence among races of the butterfly Heliconius erato inferred from patterns of mitochondrial DNA evolution" (PNAS, 91: 6491-6495) and Hu et al. (2018) “Evolutionary history of the scorpionfly Dicerapanorpa magna (Mecoptera, Panorpidae)” (Zoologica Scripta, 48: 93-105);
2. A Yule model may be more suitable for the BEAST analysis than a birth-death model;
3. Line 55, please insert a comma “,” between Hagen 1866;
4. Line 58, please add a space prior to “Field”;
5. Line 73, please change “to” into “with”. In this context, “with” should be used in combination with “compared”;
6. References: the authors used different format for different references as for colon “:” or comma “,” after the issue number of journals;
7. Penny (1977) is repeated between Lines 442 and 448, please delete Lines 442-443.

Reviewer 3 ·

Basic reporting

1. About the authors' response on my previous comment 3 contain some interesting information, that is, Kreithner (2001) did not find B. hyemalis in Austria, probably because B. hyemalis is not a common species, and it is not surprising that some previous records of B. westwoodi from Austria are misidentifications of B. hyemalis. I suggest the authors to add the information to discussion (around Line 242). The new information may give readers chances to re-check their previous Boreus records and specimens (even though in the next paragraph they mention the possibility of existence of more species in Austria).

2. The revised legend for Figure 1 (d) should be "pointed TA of B. hyemalis (d) from Austria (indicated by ...)".

3. In reference part, please check the format of DOIs of Huemer et al. (2019) and Raupach et al. (2016). They are not represented by URL.

Experimental design

No more comments.

Validity of the findings

No more comments.

Additional comments

I think the manuscript is revised well and only I mentioned a few minor points as above.

---

## Round 0.3 · accepted · Accept

Dear Dr. Zangl and colleagues:

Thanks for revising your manuscript based on the concerns raised by the reviewers. I now believe that your manuscript is suitable for publication. Congratulations! I look forward to seeing this work in print, and I anticipate it being an important resource for groups studying snow scorpionfly phylogenetics and evolution. Thanks again for choosing PeerJ to publish such important work.

Best,

-joe